# Exploring the causes of variability in quality of oropharyngeal swab sampling for SARS-CoV-2 nucleic acid testing and proposed improvement measures: a multicenter, double-blind study

Jie Zhu,[1] Fanfan Xing,[2] Yunzhu Li,[3] Chunchen Wu,[4] Shasha Li,[5] Qin Wang,[6] Jinyue Huang,[2] Yafei Zhang,[1] Xiaowei Zheng,[7] Zhenjun Liu,[8] Jianguo Rao,[9] Rui Hong,[10] Shuilin Tian,[11] Shuangyun Xiong,[12] Lin Tan,[5] Xinlei Chen,[1] Yanwu Li,[1] Wei He,[11] Xiaodan Hong,[1] Jianbo Xia,[4] Qiang Zhou,[6] Zhenhua Zhang[1]

**ABSTRACT**    Although coronavirus disease 2019 (COVID-19) has not been considered a public health emergency of international concern since last year, intermittent regional impacts still persist, and accurate testing remains crucial. Ribonuclease *P* protein subunit P30 (*RPP30*) RNA, known for its broad and stable expression in tissue cells, was used to evaluate samples from 10 hospitals with over 3,000 negative nucleic acid tests. The results revealed that the overall pass rate for the collected samples was consistently low and exhibited significant heterogeneity. After reassessing the evaluative effectiveness of *RPP30* RNA Ct values from the samples of 132 positive individuals under quarantine observation, it was used to identify factors affecting sampling quality. These factors included different stages ranging from sample collection to PCR processing, various characteristics of both samplers and individuals being sampled, as well as sampling season and location. The results indicated that post-sampling handling had minimal impact, winter and fever clinic samples showed higher quality, whereas children's samples had lower quality. The key finding was that the characteristics of samplers were closely related to sampling quality, emphasizing the role of subjectivity. Quality control warnings led to substantial improvements, confirming this finding. Consequently, although there are various factors during the testing process, the most critical aspect is to improve, supervise, and maintain standardized practices among sampling staff.

**IMPORTANCE**    This study further confirmed the reliability of internal references (IRs) in assessing sample quality, and utilized a large sample IR data to comprehensively and multidimensionally identify significant interference factors in nucleic acid test results. By further reminding and intervening in the subjective practices of specimen collectors, good results could be achieved.

**KEYWORDS**    SARS-CoV-2, PCR, *RPP30* internal reference, sampling quality

## Impact of COVID-19

The global population has been significantly impacted by the coronavirus disease 2019 (COVID-19) pandemic, which is caused by the severe acute respiratory syndrome coronavirus 2 (SARS-CoV-2) virus, commonly known as the coronavirus (1–3). In January 2021, the weekly mortality rate from COVID-19 peaked at 101,600 deaths globally (4). On 5 May 2023, the World Health Organization declared that the COVID-19 pandemic was no longer an "international public health emergency of concern." However, the limited vaccine efficacy, along with the virus's high transmissibility and

Address correspondence to Zhenhua Zhang, zzh1974cn@163.com, Qiang Zhou, zhouqiang1973@163.com, or Jianbo Xia, xjb915@126.com.

Jie Zhu, Fanfan Xing, and Yunzhu Li contributed equally to this article. Author order was determined by type of their contributions.

The authors declare no conflict of interest.

mutation rate, has led to recurrent outbreaks in many regions worldwide (5). These outbreaks continue to pose a serious risk of morbidity and mortality, particularly among the elderly and immunocompromised individuals (6, 7). Additionally, the post-acute sequelae of COVID-19, commonly known as long COVID, remains a prominent concern. In conclusion, COVID-19 still poses significant hazards and uncertainties to public health at present (8).

## Challenges in testing

Nucleic acid testing for SARS-CoV-2 is the most commonly used and accurate method for diagnosing SARS-CoV-2 infection (9, 10). The commercial test kits currently in use mainly target two gene segments: one typically targeting the conserved region of the ORF1ab gene, whereas the other commonly targets the N region or E region (11). Many test kits include pseudo-virus or plasmid samples as positive controls (12). However, due to significant differences in economics, human resources, testing technologies, and government policies among different countries or regions, a considerable proportion of test results had poor stability and low reliability, leading to the presence of false-negative phenomena (13). Furthermore, within the clinical course of the same patient, positive and negative test results at different time points may exhibit mutual contradictions (14). Sudden shifts from positive to negative and back to positive results, known as "repeated positivity," lack a clear explanation. These changes have raised concerns about possible virus reactivation or reinfection in previously recovered patients (15, 16). Given the multiple steps involved from sample collection to quantitative polymerase chain reaction (qPCR), various factors affecting the results continue to be the subject of much debate and must be comprehensively considered. Studies have suggested that different PCR test kits and virus extraction and storage reagents may have an impact on the results (17, 18). Additionally, limited studies have found that differences in the sample collection process may also have a significant impact on the test results (19, 20). Among these, a considerable proportion may be attributed to inadequate sample collection quality, which also contributes to the occurrence of "false-negative" cases (21), as infectious viral particles are primarily found extracellularly while viral nucleic acid segments are predominantly located within cells. Given the considerably higher abundance of nucleic acid segments compared to viral particles, and the observed proportional relationship between cell count and nucleic acid segments, routine PCR testing primarily targets the nucleic acid segments, which underscores the importance of ensuring an adequate cellular content in the samples (22). Currently, COVID-19 and other virus test kits, such as monkeypox virus, herpes simplex virus, parechovirus, etc., commonly utilize positive and negative controls to ensure experimental accuracy (17, 23–25). However, these methods cannot verify the sample collection quality, such as whether there is a sufficient quantity of epithelial cells in the sample. To address this, some manufacturers have developed test kits with internal references (IRs). These are a stable genetic marker stably expressed in epithelial cells. IRs allow for the assessment of cellular content in samples, ensuring that samples have adequate cell counts to reduce the likelihood of unreliable negative results (26, 27).

## *RPP30* as a sample quality IR

*RPP30*, short for ribonuclease *P* protein subunit P30, is located on human chromosome 10 (10q23.31). Its primary functions include catalysis, nuclear localization, assembly, and/or regulation of holoenzyme activity (28). As a housekeeping gene, *RPP30* exhibits high conservation, stable expression across various human tissues, and high amplification efficiency, and is unaffected by swabs and methodology. Studies have reported its use as an internal reference gene to analyze the feasibility of HIV DNA detection in cerebrospinal fluid (CSF) (29) and in the direct quantitation of SARS-CoV-2 by droplet digital PCR (30). In our previous study, our research group also demonstrated the effectiveness of the novel reference gene, *RPP30*, in accurately assessing cellular content of samples, thereby distinguishing false-negative results, which has proven to be highly

beneficial. The Ct values of *RPP30* RNA less than 25.53 and 27.48 were the optimal thresholds for avoiding false-negative ORF 1ab and N region in throat swab samples, respectively (31).

## Study objectives

This study further confirmed the reliability of using *RPP30* as an IR for assessing sample quality, and uncovered defects related to sample quality across large multicenter data sets. Through an innovative examination of various stages in the PCR process, we identified the key contributing factor to the defect—the sampling issues of the operators. Furthermore, we recommended effective corrective measures.

## MATERIALS AND METHODS

### Sample origin and features

Parameters for calculating the minimum sample size: a two-sided significance level α of 0.05, power 1-β = 0.8, and medium effect sizes according to Cohen's standards, i.e., two-sample *t*-test with d = 0.5, one-way analysis of variance (ANOVA) with f = 0.25, and linear regression with $f^2$ = 0.15. If any *RPP30* internal reference index results are missing from the samples, they will be directly excluded.

To observe the overall quality of nucleic acid samples and evaluate various factors that may affect sample quality during the sample collection process, the first set of data was collected from the samples of 10 hospitals that were not tested positive for SARS-CoV-2. Characteristics of all samples were recorded, including information about the samplers such as salary and job characteristic, sampling time, location, and method, as well as the age and gender of the individuals being sampled. In addition, to comprehensively assess the overall effect with the maximum sample size, a pooled analysis was conducted to evaluate the quality differences between hospitals or among different samplers.

In order to revalidate the reliability of *RPP30* as an IR in nucleic acid detection kits for evaluating sampling quality, the second set of data was collected from individuals with positive COVID-19 cases undergoing isolation observation at another hospital. These patients underwent nucleic acid testing every 2 days during their isolation periods. Two different test kits, "ShuoShi" (China) and "BoJie" (China), were used for each sample, both of which included N and ORF1ab segments of SARS-CoV-2 and *RPP30* IR. A Ct value greater than 38 or no Ct value was considered negative.

In cases where there is an inconsistency in the qualitative results between the two test kits and/or the two gene segments, a resampling and retest are conducted to verify the accuracy of the previous results. Additionally, if a single negative result occurs between two consecutive positive results during quarantine observation, it is considered a false negative. A retrospective analysis of these monitoring data was performed to obtain relevant data on the false-negative results. Both single and double false negatives for the two gene segments were included in the statistics. Subsequently, a correlation analysis was conducted on the Ct values between the two gene segments. The receiver operating characteristic (ROC) analysis, utilizing *RPP30* RNA Ct values, was employed to establish the criterion (Cr) value for identifying false-negative outcomes. Furthermore, a comparison was made between the *RPP30* RNA Ct values of the ORF1ab and N regions on an overall basis to assess any differences. The data collection and research workflow design are presented in Figure 1. The sample collection process was conducted in a randomized, double-blind mode to ensure that it is conducted under routine working conditions and environments. Each unit randomly selected sample test results from different time points and departments to further refine the previously mentioned sample information. Both the samplers and patients were part of the general working population, the two parties are strangers, and the sampling process does not discriminate. Additionally, both parties signed a limited disclosure informed consent form, meaning,

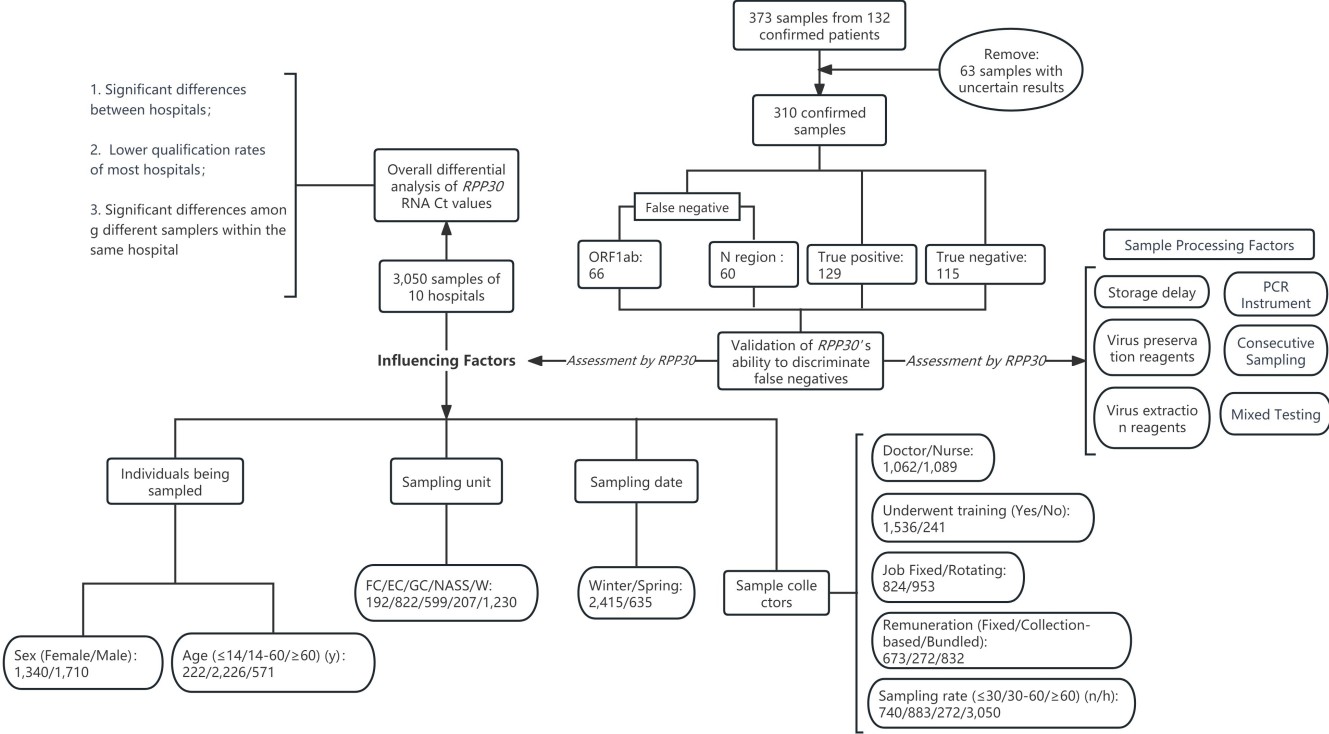

FIG 1  Research design flowchart and sample distribution. FC, fever clinic; EC, emergency clinic; GC, general clinic; NASS, nucleic acid sampling station; W, ward; y, years.

their information may be collected, but they are not aware of the specific details or purpose of the study.

## Interference in intermediate stages from sampling to detection

To investigate whether intermediate stages between sample collection and on-machine testing could potentially interfere with nucleic acid test results, we conducted multiple control experiments. These experiments encompassed factors such as the storage duration of samples prior to sequencing, different virus preservation solutions, various viral extraction reagent kits, and PCR instruments. Each experiment involved dividing the samples into *n* equal portions while ensuring that other experimental conditions remained consistent across all divisions. Because sample heterogeneity has been excluded, the final step is to determine whether different factors interfered with the results based on the Ct value of *RPP30* in each sample's PCR result.

The PCR instruments utilized encompassed three types: "Bioer" (FQD-96C, China), "Roche" (LightCycler480, USA), and "ABI" (ABI7500, USA). For virus preservation solutions, we employed three options: "Tianlong" (China), "Weimi" (China), and "ZY-Huiji" (China). In terms of viral extraction reagents, we used four varieties: "HaierShi" (China), "COYOTE" (China), "ABT" (USA), and "ZhongZhi" (China).

## Special sample collection methods

Given the variations in sample collection methods and screening frequencies across different regions and departments during nucleic acid screening, and considering that some areas have adopted pooled testing to enhance efficiency and reduce costs, we conducted two experiments to compare single testing with different levels of sample pooling and detected the samples collected from volunteers who underwent four consecutive samplings within a span of 1 hour. Following the same logic, while keeping other potential interfering factors consistent, the Ct value of *RPP30* in the PCR results was

used to determine whether there are differences between the two sample processing methods.

## Influence of different sampler characteristics on sampling quality

After addressing the potential confounding effects of the key stages in nucleic acid testing mentioned above, we further focused on the impact of sampler-related characteristics on sample quality. A further analysis was conducted using linear regression to assess the influence of sampler characteristics on sampling quality. This analysis encompassed factors such as their occupation, salary structure, work arrangement, hourly sampling rate, and whether they underwent training. Data filtering ensures that each sample possesses complete feature information. Furthermore, to ensure the reliability of this analysis, factors identified in the earlier stages of analysis that had a disruptive effect on the results were normalized. For instance, spring season sampling samples and samples from children were excluded.

## Value of quality control alerts in improving sampling quality

To confirm that there is room for improvement in the samplers' work quality, we chose a hospital's centralized nucleic acid collection site and gathered a specific amount of reference data from samples collected on a typical business day. Another set of sample data was sourced from samples collected by sampling personnel who were subjected to quality control warnings before beginning their work. They received written and verbal notifications from the hospital-level leadership, which included both rewards and penalties based on the sample quality assessment results. Specific warning details include: (i) the addition of a quality control system for COVID-19 nucleic acid testing results; (ii) a retrospective analysis of past samples revealed a significant proportion of non-compliant samples; (iii) each sample can be traced back to the individual who collected it; and (iv) in the future, quality control of sampling will be strengthened, and individuals whose samples exceed a 10% non-compliance rate after sampling analysis will receive a warning and a 20% deduction from their salary for the corresponding work period. The differences in Ct values between these two sets of data were compared to assess the impact of the warnings.

## Statistical analysis

Statistical analysis was performed using SPSS version 25.0, and visualization was carried out via GraphPad Prism version 8.0. For continuous variables following a normal distribution or when the sample size was >200, we used mean ± standard deviation, $t$-tests for comparisons between the two groups, and one-way ANOVA for comparing the groups. Otherwise, median ± fourth quartile and Mann-Whitney U or Kruskal-Wallis tests were used. Fisher's exact test was used to compare rates between the groups. Because the stability of sampling quality is crucial in the COVID-19 screening process, we introduced the coefficient of variation (CV) for Ct values, calculated as CV = (standard deviation / mean) × 100%. The results were considered statistically significant if the two-tailed $P$ is <0.05.

## RESULTS

### Observing the overall sampling quality of various hospitals using IR Ct values

A total of 3,050 nucleic acid samples were collected from 10 hospitals, and the aforementioned sample characteristics were comprehensively documented. Due to the diversity of data sources and the extended time span, some samples lack precise information on specific characteristics, resulting in uneven total sample sizes for various attributes (Table 1).

By comparing the *RPP30* RNA Ct values of the samples, it was observed that the overall sampling qualities of most hospitals were subpar. If the Ct values of 25.53 (red

TABLE 1 Overall characteristics and *RPP30* levels of the samples from 10 hospitals[a]

| | Number | Ct values (mean ± SD) |
|---|---|---|
| **Individuals being sampled** | | |
| Sex | | |
| Female | 1,340 | 28.19 ± 2.88 |
| Male | 1,710 | 28.07 ± 2.85 |
| Age (years) | | |
| ≤14 | 222 | 28.91 ± 2.60 |
| 14–60 | 2,226 | 28.11 ± 2.86 |
| ≥60 | 571 | 28.01 ± 2.96 |
| Sampling unit | | |
| Fever clinic | 192 | 27.76 ± 1.84 |
| Emergency clinic | 822 | 29.00 ± 1.92 |
| General clinic | 599 | 28.98 ± 2.43 |
| Nucleic acid sampling station | 207 | 29.16 ± 2.75 |
| Ward | 1,230 | 28.77 ± 2.54 |
| Sampling date | | |
| Winter (Dec.–Feb.) | 2,415 | 28.05 ± 2.89 |
| Spring (Apr.–May) | 635 | 29.43 ± 2.17 |
| **Sample collectors** | | |
| Occupation | | |
| Doctor | 1,062 | 27.81 ± 2.92 |
| Nurse | 1,089 | 28.14 ± 2.66 |
| Underwent training | | |
| Yes | 1,536 | 28.29 ± 2.66 |
| No | 241 | 29.16 ± 1.00 |
| Job modes | | |
| Fixed | 824 | 29.51 ± 2.58 |
| Rotating | 953 | 27.46 ± 2.02 |
| Remuneration methods | | |
| Fixed | 673 | 28.86 ± 2.42 |
| Collection based | 272 | 29.32 ± 1.25 |
| Bundled | 832 | 26.22 ± 2.28 |
| Sampling rate (n/h) | | |
| ≤30 | 740 | 29.16 ± 1.02 |
| 30–60 | 883 | 28.16 ± 2.39 |
| ≥60 | 272 | 28.22 ± 2.72 |
| **Total** | 3,050 | 28.12 ± 2.87 |

[a]SD, standard deviation; n, number; h, hour.

line) and 27.48 (blue line) were considered to be good and acceptable sampling quality thresholds, respectively, the compliance rates of majority of hospitals were relatively low (Fig. 2A). Additionally, there was a significant heterogeneity that exists between different hospitals (Fig. 2A), and most pairwise comparisons, including *RPP30* Ct values (Fig. 2B) and ratios of *RPP30* Ct values <25.53 (Fig. 2C), indicated significant differences. Furthermore, two hospitals were selected based on their relatively low (Fig. 2D) and high (Fig. 2E) overall *RPP30* Ct values, and there were still significant differences in the detection results of different samplers between the two hospitals.

## Verifying the effectiveness of *RPP30* in evaluating sampling quality

Based on previous research findings, IR Ct values below 25.53 effectively avoid false-negative results of ORF 1ab and N regions (31). We collected a total of 373 samples from 132 COVID-19 patients at the quarantine sites, and deleted negative samples that received only one test or of which the results were uncertain. Using the aforementioned

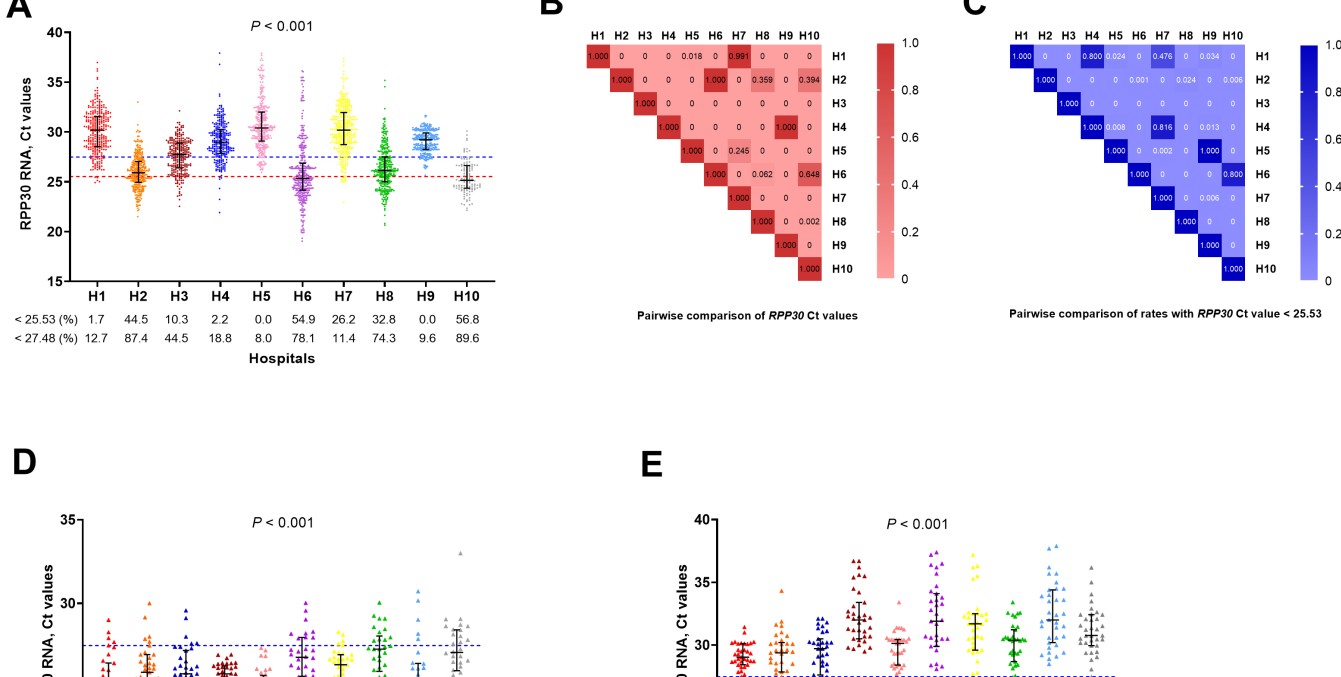

**FIG 2** Comparison of *RPP30* RNA Ct values among 10 hospitals. (A) Sampling quality was severely inadequate and inconsistent among multiple hospitals (*P* < 0.001); (B) pairwise comparison of *RPP30* Ct values; (C) pairwise comparison of the rates with *RPP30* Ct values <25.53; (D) comparison of *RPP30* results for samplers from H2; (E) comparison of *RPP30* results for samplers from H5 (red line: value = 25.53; blue line: value = 27.48; H, hospital; S, sampler; light red/blue to deep red/blue: the *P* value gradually increases from 0 to 1).

method, we identified 129 true positives and 115 true negatives for both genes, 60 false negatives for the N gene, and 66 false negatives for the ORF1ab gene in 310 confirmed positive or negative samples.

By analyzing the test results of positive samples, a strong correlation, as expected, between the Ct values of the ORF1ab and N regions was revealed (*r* = 0.977, *P* < 0.001) (Fig. 3A). Additionally, the Ct values of the ORF1ab (*r* = 0.788, *P* < 0.001) (Fig. 3B) and N (*r* = 0.776, *P* < 0.001) (Fig. 3C) regions both showed a significant correlation with the Ct values of *RPP30* RNA. Moreover, *RPP30* RNA exhibited a strong discriminative capability for both the ORF1ab (AUC: 0.968, *P* < 0.001) (Fig. 3D) and N (AUC: 0.964, *P* < 0.001) (Fig. 3E) regions in distinguishing false-negative results. The Ct values used in previous studies to identify the false negatives of ORF1ab (25.53) and N (27.48) were applied respectively, and these samples also had high sensitivity (ORF1ab: 0.955, N: 0.933) and specificity (ORF1ab: 0.815, N: 0.915) (Fig. 3D and E). Among the 310 confirmed samples, ORF1ab had the lowest proportion of single false-negative cases (7.1%), whereas the combination of double false-negative cases and confirmed negative samples reached 49.35% (Fig. 3F). Furthermore, the false-negative *RPP30* Ct values of the ORF1ab and N regions were significantly higher than the corresponding true-positive samples (*P* < 0.001) (Fig. 3G).

## Interference effects in the specimen handling process

Through various experiments, short-term storage delays before testing (Fig. 4A), different virus preservation reagents (Fig. 4B), different virus extraction reagents (Fig. 4C), different PCR instruments (Fig. 4D), continuous sampling of the same individual in a short period

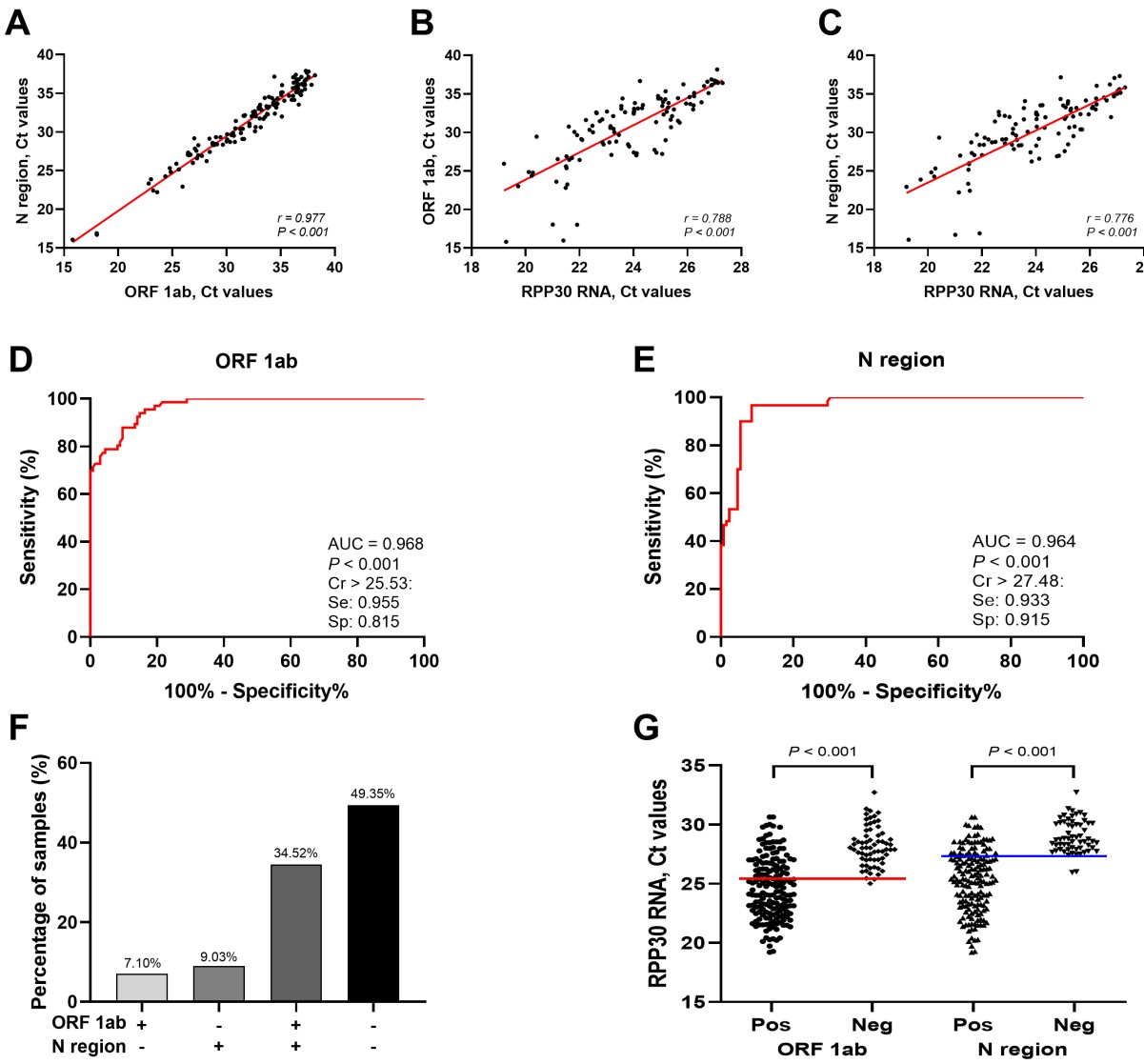

**FIG 3** The reliability of using *RPP30* RNA Ct value to evaluate the testing quality of the SARS-CoV-2. (A) Correlation between ORF 1ab and N region Ct values (*r* = 0.977, *P* < 0.001); (B and C) correlation of Ct values of ORF1ab (*r* = 0.788, *P* < 0.001) and N (*r* = 0.776, *P* < 0.001) regions with that of *RPP30*. ROC curve and AUC were calculated for the Ct values of *RPP30* to predict the false-negative results of ORF 1ab (D) (AUC = 0.968, *P* < 0.001) and N (E) (AUC = 0.964, *P* < 0.001) regions from positive samples. (F) Proportion of single positive, double positive, and double negative of ORF 1ab and N regions in 310 samples. (G) Ct values of *RPP30* corresponding to positive and negative samples of ORF1ab and N regions; red line = 25.53, blue line = 27.48. ROC, receiver operating characteristic; AUC, area under the curve; Cr, criterion (cutoff value); Se, sensitivity; Sp, specificity; Pos, positive; Neg, negative.

(Fig. 4E), and single-tube single testing or single-tube mixing below 10 mixes (Fig. 4F) were observed, which did not significantly affect the test results (*P* > 0.05).

## Influence of non-sampler factors on test results

A univariate analysis was conducted to evaluate the impact of characteristics of the sampled population, sampling season, and collection department on the IR Ct values of test results. It was observed that the gender of the sampled individuals did not have any impact on *RPP30* RNA Ct values (*P* > 0.05) (Fig. 5A). The Ct values of pediatric samples were significantly higher than that of adults and elderly people (both *P* < 0.001) (Fig. 5B). The Ct values of spring samples were significantly higher than those of winter samples (*P* < 0.001) (Fig. 5C). The Ct values of samples collected from fever clinic were significantly lower than those of samples from other departments (*P* < 0.001), whereas no significant

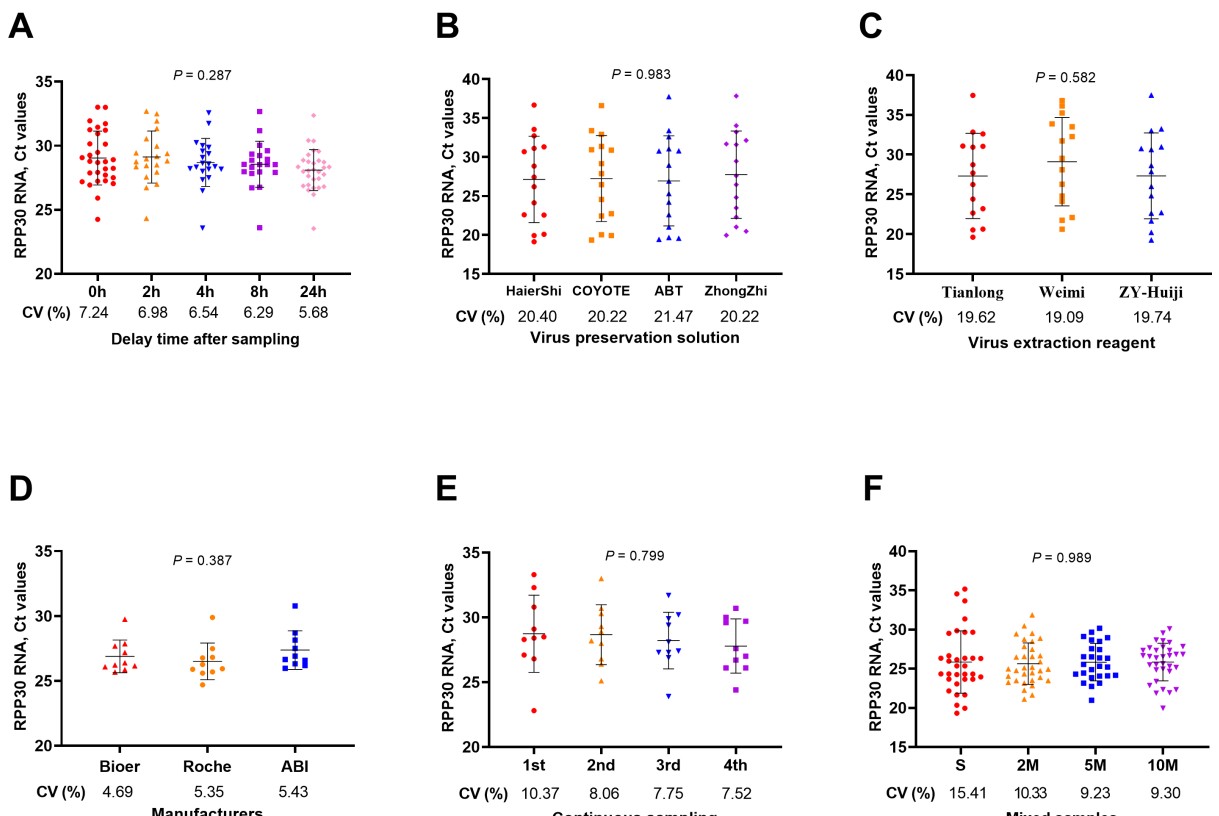

**FIG 4** Differences in sample quality under various sample processing conditions. Variations in different processing conditions, such as delayed time after sampling (A), virus preservation solution (B), virus extraction reagents (C), different PCR instruments (D), consecutive sampling (E), and mixed sampling (F), did not result in significant differences in *RPP30* RNA Ct values (*P* > 0.05). CV, coefficient of variation; S, single sample test; M, mixed sample test; PCR, polymerase chain reaction.

difference of the Ct values was observed between samples from other departments (*P* > 0.05) (Fig. 5D).

## Impact of individual factors of the sample collectors on sampling quality

A total of 1,146 samples were collected with comprehensive information on the sample collectors (Table 2). Using the *RPP30* RNA Ct value as the outcome variable, a linear regression analysis was conducted to examine the differences in various characteristics of the sample collectors. The results indicated that the IR Ct values of samples collected by doctors were significantly lower than those collected by nurses. The shift work system was more effective in reducing the Ct values compared to a fixed one. Among the three compensation methods, the "bundled payment" method improved the IR Ct values. The Ct values were the lowest when the sampling speed was 30–60 samples per hour, whereas excessively fast sampling or methods based on the number of samples collected seemed to compromise the sampling quality. Furthermore, although sampling training did not significantly reduce the IR Ct values, the overall Ct values were lower. This may be related to the varying quality and strictness of training across different units (Fig. 6A).

## The role of quality control warnings for sample collectors

Comparison of the IR Ct values of 238 samples collected without quality control warning and 210 samples with warnings revealed that the IR Ct values of the latter were significantly lower (*P* < 0.001). The CV notably decreased from 12.64% to 9.15% after

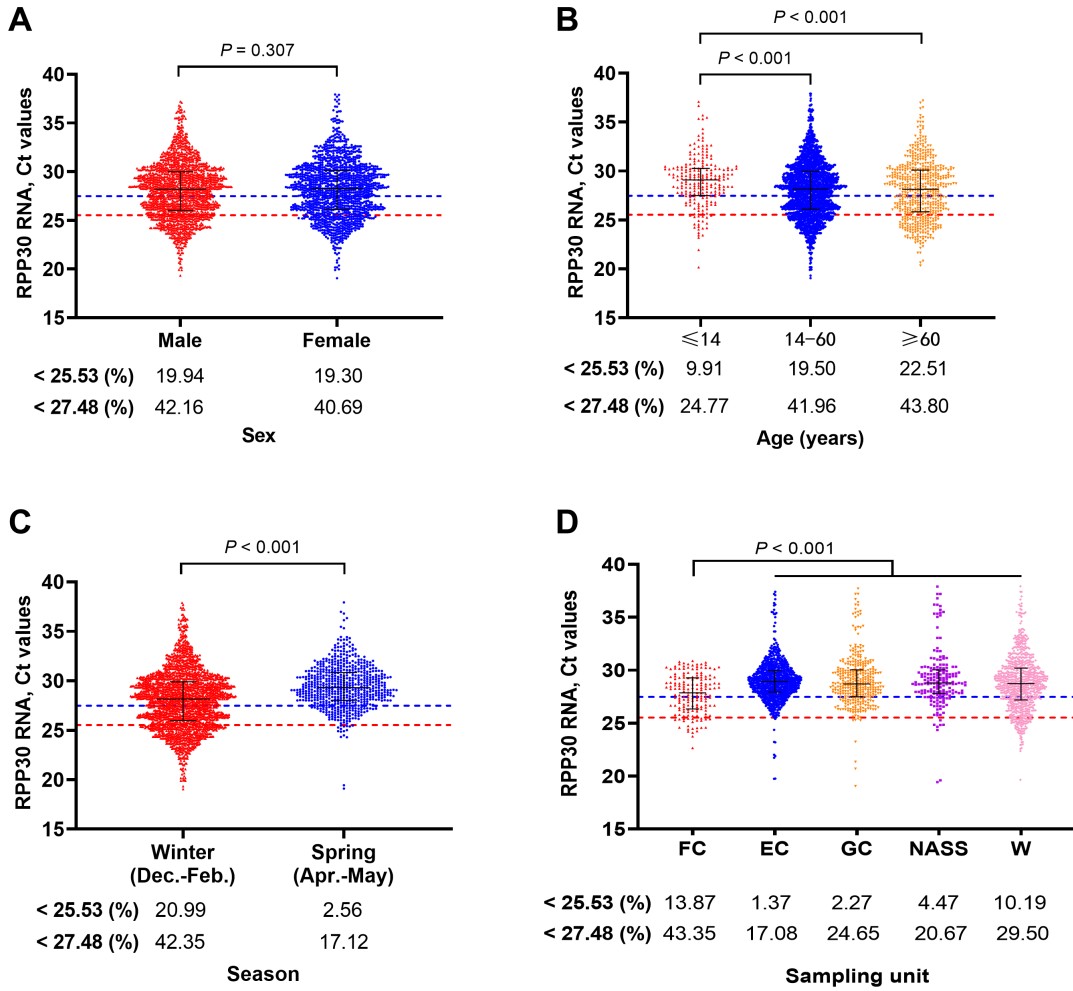

**FIG 5** Assessment of the impact of differences in sampling characteristics on *RPP30* RNA. (A) Influenced by the gender of the participants ($P > 0.05$); (B) the *RPP30* RNA Ct values of samples from pediatric participants were significantly higher than those from other age groups ($P < 0.001$); (C) the *RPP30* RNA Ct values of winter samples were significantly lower than those of spring samples ($P < 0.001$); (D) samples from the fever clinic exhibit significantly lower *RPP30* RNA Ct values compared to other sampling departments ($P < 0.001$), whereas no significant differences were observed among the other units ($P > 0.05$). Red line = 25.53, blue line = 27.48; FC, fever clinic; EC, emergency clinic; GC, general clinic; NASS, nucleic acid sampling station; W, ward.

quality control warning. In addition, the proportion of the *RPP30* Ct values lower than 25.53 and 27.48 increased from 51.26% to 71.9% and from 71.85% to 91.43%, respectively (Fig. 6B).

## DISCUSSION

Accurately and rapidly identifying infected individuals remains a crucial measure to curb the spread of SARS-CoV-2. Real-time reverse transcription-polymerase chain reaction (RT-PCR) is a widely used method for the detection of SARS-CoV-2 nucleic acid, which has the advantages of short turnaround time, convenience, strong specificity, and good reproducibility (11). PCR testing encompasses both intracellular nucleic acid fragments and extracellular viral particles, with the former being significantly more abundant than the latter. Nucleic acid segments can persist for a long period of time, whereas infectious viral particles have a short lifespan and typically disappear within 5–7 days after the symptom onset. The quantity of nucleic acid segments is directly proportional to the number of cells, with *RPP30* serving as the cellular reference marker. Previous studies have also confirmed the Ct threshold value for *RPP30* (31). Therefore, *RPP30* exhibits a positive correlation with nucleic acid segments and can be used as an internal control

**TABLE 2** Samples with complete sampler information were used for linear regression analysis[a]

| Sample collector features | Number | Ct values (mean ± SD) |
|---|---|---|
| Occupation | | |
| Doctor | 250 | 26.14 ± 2.32 |
| Nurse | 896 | 28.55 ± 2.47 |
| Underwent training | | |
| Yes | 995 | 28.23 ± 2.02 |
| No | 151 | 29.12 ± 1.76 |
| Job modes | | |
| Fixed | 433 | 29.95 ± 2.49 |
| Rotating | 713 | 27.38 ± 2.07 |
| Remuneration methods | | |
| Fixed | 587 | 28.66 ± 2.36 |
| Collection based | 288 | 29.28 ± 1.51 |
| Bundled | 271 | 26.15 ± 2.27 |
| Sampling rate (n/h) | | |
| ≤30 | 180 | 29.18 ± 1.25 |
| 30–60 | 664 | 27.56 ± 2.37 |
| ≥60 | 302 | 28.44 ± 2.52 |
| **Total** | 1,146 | 28.35 ± 2.78 |

[a]SD, standard deviation; n, number; h, hour.

for sampling quality in individuals with respiratory tract infection. In addition, current assay kits typically target two or more viral regions to ensure specificity and accuracy of test results, including the conservatively specific ORF1ab region. Nevertheless, the operational reality continues to witness the emergence of significant false-negative results (32).

In this study, we found significant differences in the *RPP30* Ct values of oropharyngeal swab samples from 10 different hospitals in terms of direct quantification, variation rate, pass rate, and excellent rate. Even within the same hospital, substantial differences were observed between different samplers. The aim of the study was to objectively describe the current situation of nucleic acid testing and to investigate the factors that have the greatest impact on this phenomenon using a multicenter, double-blind research method.

The viral load of SARS-CoV-2 varies from high to low in the following order: bronchoalveolar lavage fluid (optimal), deep cough sputum, nasopharyngeal swab, oropharyngeal swab, and blood (33, 34). In spite of this, oropharyngeal and/or nasopharyngeal swabs are utilized as the routine type of specimen, whereas samples such as bronchoalveolar lavage fluid, which could enhance detection rates, face challenges in widespread implementation. This underscores the need for effective assessment and control of upper respiratory tract sample collection quality. With multiple steps involved in the nucleic acid testing process, each juncture could potentially influence results (35, 36).

The data of the study encompassed various characteristics of more than 3,000 samples from multiple hospitals, including sampled individuals, collectors, sampling methods, and sample processing. The survey results showed that most of the hospitals had a high rate of sampling quality failure, and the quality varied greatly among different hospitals. The feasibility of using 25.53 as the cutoff value to identify sampling quality was verified again (31).

Although there are some differences in the correlation between viral nucleic acid fragments and *RPP30*, as well as the ROC analysis results for single and dual false-negative cases, compared to our previous findings, we have analyzed the underlying reasons. The selected samples included persons with mild or asymptomatic infection who were under quarantine. Their viral loads were inherently low, and there was a tendency for viral levels to gradually decrease on repeated testing during the quarantine period.

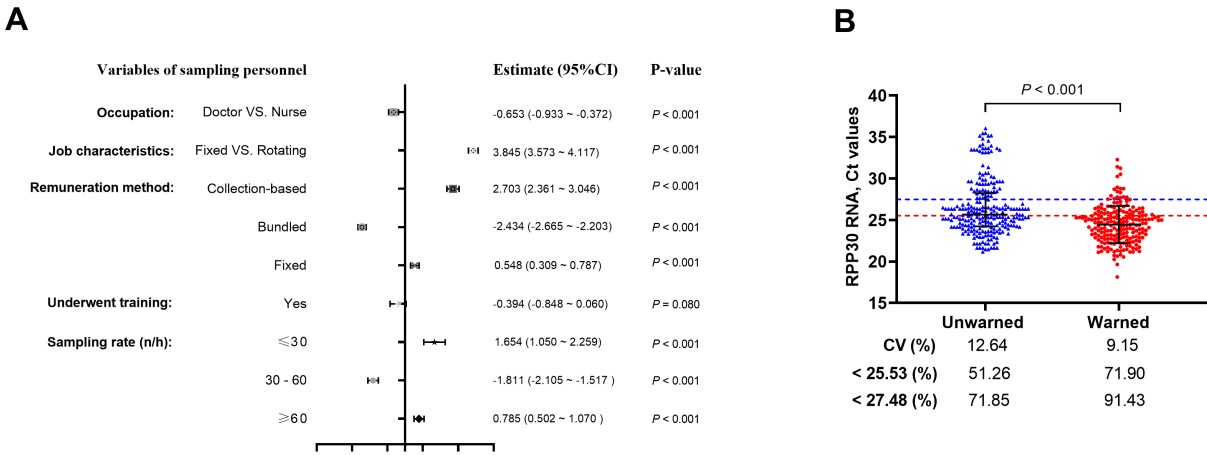

**FIG 6** Linear regression and improvement strategies for the influence of sampler characteristics on *RPP30* RNA Ct values. (A) Sampling characteristics such as doctor sampling, rotational shift schedules, bundled compensation method, and appropriate sampling rates can significantly reduce the Ct values of the IR in the samples (*P* < 0.001). (B) Quality control warnings significantly improve sampling quality; CV and the rate below the two thresholds were significantly increased. CV, coefficient of variation; *P* < 0.001.

This phenomenon might have affected the correlation between the IR and nucleic acid Ct values, as well as the establishment of the cutoff value. This method has a greater advantage in distinguishing between weakly positive and negative samples.

By analyzing the IR Ct values, we intensively examined the factors that affected the quality of sample collection. During the nucleic acid screening, certain populations might undergo multiple consecutive samplings within a short period of time. We conducted four consecutive samples of a group of 10 volunteers within an hour. Interestingly, no significant difference in *RPP30* Ct values was observed in these samples. Given the large sample quantity collected and processed daily, some lag in the processing of samples from collection to analysis is inevitable. Nonetheless, controlled experimental studies have shown that there was no significant deviation from the test results within 24 hours of routine storage. For the mixed sampling method commonly used in nucleic acid screening, we compared the IR Ct values of single and mixed testing at ratios of 2, 5, and 10, and the results showed no statistically significant difference. However, it should be noted that the total number of epithelial cells in a pooled sample should be higher than in a single sample theoretically. Despite this, the IR Ct values did not increase with increasing levels of mixing, indicating the need to avoid excessive mixing. This phenomenon might be caused by repeated opening of the sample tubes and manual handling. Furthermore, we conducted a comprehensive prospective study on the factors that may influence test results at various stages from sampling to analysis. Although some studies have shown that variations in consumables can affect test results (37), our overall experimental results indicated that relatively objective factors, such as differences in assay kits, reagents, and PCR instruments, did not result in significant differences in test results. This suggests the maturity and reliability of commercial reagents after prolonged exposure to the epidemic, and underscores the stability of PCR instruments as a widely used routine testing modality (38).

Among the characteristics of the sampled population, only minors had a relatively low sample quality, possibly due to their lack of tolerance and cooperation in the sampling process. Interestingly, samples collected during winter exhibited significantly higher quality compared to those collected during spring. This discrepancy may arise from multiple factors. Firstly, the higher incidence of COVID-19 during winter may lead to greater vigilance among sample collectors. Additionally, studies have confirmed that temperature affects the stability of nucleic acid samples (39). Further investigation is warranted to determine the extent to which seasonal variations are influenced by external temperature, as well as the specific stages at which temperature affects them.

On the whole, objective factors had relatively little effect on sample quality. The main reasons for the inadequate sampling quality were the technical competence and subjective initiative of samplers, which was consistent with the focus of a meta-analysis study (40). Employing linear regression analysis, we evaluated the impact of various collector characteristics on sampling quality. It was discovered that work intensity, represented by sampling rate, nature of employment, and compensation methods, significantly affected the sampling quality. This indicates that the subjective attitude of sample collectors plays a critical role in determining sampling quality. Although the presence or absence of training did not yield significant differences in sample results, the importance of training could not be denied. The need to improve the quality of training had been indicated by previous studies (41). A sampling rate at 30–60 samples per hour demonstrated an optimal sampling quality. This may be due to the fast rate of sampling, which makes it difficult to continuously collect a sufficient number of epithelial cells. On the contrary, slower sampling rates may be related to individuals not being familiar with sampling techniques.

To underscore the importance of enhancing the attitudes and sense of responsibility among staff, we had introduced pre-sampling warnings for sample collectors. These warnings served as notification of upcoming quality monitoring of their sampling tasks, as well as the possibility of penalties for non-compliance. Afterward, a comparative analysis was carried out on the sampling quality within the same group of collectors pre- and post-warnings, revealing a noteworthy enhancement following the warning. This emphasizes the importance of fostering positive attitudes, fostering standardized work ethics, and instilling a sense of responsibility in sample collectors. Furthermore, it underscores the significance of effectively managing workloads and decoupling compensation from the quantity of samples collected, thus reducing the prevalence of non-compliant samples on a large scale. However, the warning measures provided in this study may not be applicable to every unit or workgroup, and units should analyze specific conditions when developing sampling improvement measures.

This study, based on multicenter random sample data, indicates that the sample quality issues observed are quite representative. After extensive screening of multiple factors in nearly all key stages of nucleic acid testing, it was found that sampler-related factors have the greatest impact, which is both stable and noteworthy. The significant improvement in sample quality following the warning not only provides valuable insights for developing improvement measures but also serves as a reverse validation of the main conclusions, reflecting the robustness of the conclusions and their potential value for broader application. However, the study has some limitations. Due to the complexity of nucleic acid testing details, it is challenging to cover all aspects, and some may even be uncontrollable or unnoticeable. Additionally, the characteristics affecting sampler work quality that we collected are relatively limited and likely include more factors than analyzed here, which may require further review by managers. Moreover, there are other factors affecting sample quality that are worth exploring, such as the impact of different transportation methods on results. Further attention and continuous improvement are needed for more aspects of nucleic acid testing staff management, including methods, techniques, and training.

In summary, given the potential long-term presence of SARS-CoV-2 as a pathogenic virus, sustained vigilance and continual enhancement of preventive, diagnostic, and therapeutic measures remain imperative.

## ACKNOWLEDGMENTS

Thanks to the more than 100 nucleic acid collection workers for their support in this study.

This study was funded by the Anhui Provincial Natural Science Foundation (Science and Technology Department of Anhui Province, 2108085MH298), the scientific research project of the Second Affiliated Hospital of Anhui Medical University (the Second Affiliated Hospital of Anhui Medical University, 2021LCZD01, 2021lcxk027).

Z.Z., Q.Z., and J.X. conceptualized and designed the study, and confirmed the final draft; J.Z. summarized and analyzed the data and drafted the initial manuscript; F.X., Y.L., and J.H. carried out some experiments and revised the manuscript; C.W., S.L., Q.W., and X.Z. were responsible for some experiments and data collection, and participated in the discussion of the project design; Z.L., J.R., R.H., S.T., S.X., L.T., Y.L., and W.H. participated in part in the data collection, coordination, and promotion of the study; X.C. assisted in data collection and participated in the discussion of the manuscript writing; Y.Z. and X.H. participated in the coordination and promotion of the research work and in the revision of the manuscript. All authors read and approved the final manuscript.

The authors declare that there is no conflict of interest regarding the publication of this paper.

## AUTHOR AFFILIATIONS

[1]Institute of Clinical Virology, Department of Infectious Diseases, The Second Affiliated Hospital of Anhui Medical University, Hefei, China

[2]Department of Clinical Microbiology and Infection Control, The University of Hong Kong-Shenzhen Hospital, Shenzhen, China

[3]Department of Infectious Diseases, The Third Affiliated Hospital of Anhui Medical University, The First People's Hospital of Hefei, Hefei, China

[4]Department of Laboratory Medicine, Maternal and Child Health Hospital of Hubei Province, Tongji Medical College, Huazhong University of Science and Technology, Wuhan, China

[5]Division of Liver Disease, The Second People's Hospital of Fuyang City, Fuyang, China

[6]Department of Clinical Laboratory, The Second Affiliated Hospital of Anhui Medical University, Hefei, China

[7]Department of Infectious Diseases, The First Affiliated Hospital of University of Science and Technology of China, Hefei, China

[8]Department of Infectious Diseases, Anqing Municipal Hospital, Anqing, China

[9]Department of Infectious Diseases, The Lu'an Affiliated Hospital of Anhui Medical University, Lu'an People's Hospital, Lu'an, China

[10]Department of Infectious Diseases, Tongling Municipal Hospital, Tongling, China

[11]Division of Liver Disease, Traditional Chinese Hospital of LuAn, Anhui University of Traditional Chinese Medicine, Lu'an, China

[12]Department of Infectious Diseases, Funan County People's Hospital, Fuyang, China

## AUTHOR ORCIDs

Fanfan Xing  https://orcid.org/0000-0001-8233-5127
Jianbo Xia  http://orcid.org/0000-0002-8912-9921
Zhenhua Zhang  http://orcid.org/0000-0002-8480-9004

## AUTHOR CONTRIBUTIONS

Jie Zhu, Formal analysis, Investigation, Methodology, Software, Validation, Visualization, Writing – original draft | Fanfan Xing, Data curation, Methodology, Resources, Writing – review and editing | Yunzhu Li, Methodology, Writing – review and editing | Chunchen Wu, Project administration, Resources, Software | Shasha Li, Data curation, Methodology, Resources | Qin Wang, Data curation, Project administration, Resources | Jinyue Huang, Formal analysis, Writing – review and editing | Yafei Zhang, Project administration, Writing – review and editing | Xiaowei Zheng, Data curation, Resources, Validation | Zhenjun Liu, Data curation | Jianguo Rao, Data curation | Rui Hong, Data curation | Shuilin Tian, Data curation, Project administration | Shuangyun Xiong, Data curation, Supervision | Lin Tan, Data curation | Xinlei Chen, Data curation, Writing – original draft | Yanwu Li, Data curation, Project administration | Wei He, Data curation | Xiaodan Hong, Writing – review and editing | Jianbo Xia, Conceptualization, Supervision, Writing – review and editing | Qiang Zhou, Conceptualization, Project administration, Supervision |

Zhenhua Zhang, Conceptualization, Project administration, Supervision, Writing – review and editing

## DATA AVAILABILITY

The data sets used and/or analyzed during the current study are available from the corresponding author on reasonable request.

## ETHICS APPROVAL

The study was approved by the Ethics Committee of Anhui Medical University (Number: 20200832).

Written informed consents were obtained from all participants.

## ADDITIONAL FILES

The following material is available online.

### Open Peer Review

**PEER REVIEW HISTORY (review-history.pdf).** An accounting of the reviewer comments and feedback.

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
