## [Reviewer comments · Microbiology Spectrum]

Microbiology Spectrum

Exploring the Causes of Variability in Quality of Oropharyngeal Swab Sampling for SARS-CoV-2 Nucleic Acid Testing and Proposed Improvement Measures: A Multicenter, Double-Blind Study

Jie Zhu, Fanfan Xing, Yunzhu Li, Chunchen Wu, Shasha Li, Qin Wang, Jinyue Huang, Yafei Zhang, Xiaowei Zheng, Zhenjun Liu, Jianguo Rao, Rui Hong, Shuilin Tian, Shuangyun Xiong, Lin Tan, Xinlei Chen, Yanwu Li, Wei He, Xiaodan Hong, Jianbo Xia, Qiang Zhou, and Zhenhua Zhang

Corresponding Author(s): Zhenhua Zhang, Second Affiliated Hospital of Anhui Medical University

Review Timeline:

Submission Date:	June 26, 2024
Editorial Decision:	August 18, 2024
Revision Received:	August 30, 2024
Editorial Decision:	September 7, 2024
Revision Received:	September 8, 2024
Accepted:	September 9, 2024

Editor: Benjamin Liu

Reviewer(s): The reviewers have opted to remain anonymous.

Transaction Report:

DOI: <https://doi.org/10.1128/spectrum.01567-24>

Re: Spectrum01567-24 (Exploring the Causes of Variability in Quality of Oropharyngeal Swab Sampling for SARS-CoV-2 Nucleic Acid Testing and Proposed Improvement Measures: A Multicenter, Double-Blind Study)

Dear Prof. Zhenhua Zhang:

Thank you for the privilege of reviewing your work. Below you will find my comments, instructions from the Spectrum editorial office, and the reviewer comments.

Editor's comments:

1. The authors need to clarify what is RPP30 and give the full term of RPP30
2. The manuscript provides confusing information that may mislead readers to assume SARS-CoV-2 PCRs are quantitative, which is actually qualitative. Please double check
3. Line 71-72: "These outbreaks continue to pose a serious risk of morbidity and mortality, particularly among the elderly and immunocompromised individuals". More references should be added, with the following as examples:

Role of Host Immune and Inflammatory Responses in COVID-19 Cases with Underlying Primary Immunodeficiency: A Review. *J Interferon Cytokine Res.* 2020 Dec;40(12):549-554. doi: 10.1089/jir.2020.0210. PMID: 33337932; PMCID: PMC7757688.

4. Line 63-64: "The global population has been significantly impacted by the COVID-19 pandemic, which is caused by the SARS-CoV-2 virus, commonly known as the coronavirus". The authors should cite more references, with the following as example:

Clinical significance of measuring serum cytokine levels as inflammatory biomarkers in adult and pediatric COVID-19 cases: A review. *Cytokine.* 2021 Jun;142:155478. doi: 10.1016/j.cyto.2021.155478. Epub 2021 Feb 23. PMID: 33667962; PMCID: PMC7901304.

The Brief Case: Ventilator-Associated *Corynebacterium accolens* Pneumonia in a Patient with Respiratory Failure Due to COVID-19. *J Clin Microbiol.* 2021 Aug 18;59(9):e0013721. doi: 10.1128/JCM.00137-21. Epub 2021 Aug 18. PMID: 34406882; PMCID: PMC8372998.

5. Line 104-107: "Currently, COVID-19 and other virus test kits commonly utilize positive and negative standards to ensure experimental accuracy. However, these methods cannot verify the sample collection quality, such as whether there is a sufficient quantity of epithelial cells in the sample." It would be better to explain what "other virus" means here and add more examples and references to support this statement. For example, mpox PCR testing heavily relies on quality samples. This is also true for HSV and human parechovirus (HPeV) testing. The authors are suggested to add this into discussion. Some suggested references are listed here for the authors to consider:

Mpox (Monkeypox) Virus and Its Co-Infection with HIV, Sexually Transmitted Infections, or Bacterial Superinfections: Double Whammy or a New Prime Culprit? *Viruses.* 2024 May 15;16(5):784. doi: 10.3390/v16050784. PMID: 38793665; PMCID: PMC11125633.

The laboratory diagnosis of herpes simplex virus infections. *Can J Infect Dis Med Microbiol.* 2005 Mar;16(2):92-8. doi: 10.1155/2005/318294. PMID: 18159535; PMCID: PMC2095011.

Optimization and evaluation of a novel real-time RT-PCR test for detection of parechovirus in cerebrospinal fluid. *J Virol Methods.* 2019 Oct;272:113690. doi: 10.1016/j.jviromet.2019.113690. Epub 2019 Jul 5. PMID: 31283959.

Please return the manuscript within 15 days; if you cannot complete the modification within this time period, please contact me. If you do not wish to modify the manuscript and prefer to submit it to another journal, notify me immediately so that the manuscript may be formally withdrawn from consideration by Spectrum.

Revision Guidelines

- Upload point-by-point responses to the issues raised by the reviewers in a file named "Response to Reviewers," NOT in your cover letter.
- Upload a compare copy of the manuscript (without figures) as a "Marked-Up Manuscript" file.
- Upload a clean .DOC/.DOCX version of the revised manuscript and remove the previous version.

- Each figure must be uploaded as a separate, editable, high-resolution file (TIFF or EPS preferred), and any multipanel figures must be assembled into one file.
- Any supplemental material intended for posting by ASM should be uploaded with their legends separate from the main manuscript. You can combine all supplemental material into one file (preferred) or split it into a maximum of 10 files with all associated legends included.

Sincerely,
Benjamin Liu
Editor
Microbiology Spectrum

Reviewer #2 (Comments for the Author):

The study is noteworthy because it is multicenter and has many cases. However, detailed information is required for false positivity. Also, I could not fully understand the ages of the patients. Was there standardization in the tests for under 2, 2-5, 5-12, 12-18 and 18 and over? I could not understand the age section. Was there a difference between lower respiratory tract Covid infection and upper respiratory tract COVID infection? Best regards.

Reviewer #3 (Comments for the Author):

INTRODUCTION

Strengths

Comprehensive Overview: The introduction provides a thorough overview of the impact of the COVID-19 pandemic, covering mortality rates, ongoing risks, and the persistence of long COVID.

Focus on Testing Issues: The section highlights the critical role of nucleic acid testing and the challenges associated with it, such as false negatives and variability in test results.

Relevance: Discusses the practical implications of testing reliability and the importance of accurate sample collection.

Areas for Improvement

Clarity and Conciseness: The introduction could benefit from more concise language. Some sentences are lengthy and could be split for better readability.

Structure and Flow: While the content is comprehensive, it can be better structured. Grouping related points together and using subheadings may enhance readability.

In-depth Explanation: Certain terms and concepts, such as 'repeated positivity' and 'internal references (IRs)', could be explained in more detail for readers who may not be familiar with these terms.

Citations and Evidence: Ensure that every claim, especially those related to the efficacy of specific genes like RPP30, is backed by citations to relevant studies or data.

Focus on Novel Contributions: The introduction should more clearly highlight the novel contributions of the study, setting it apart from existing literature.

Specific Suggestions

Simplify Language: For example, instead of "During the initial three years of the COVID-19 pandemic, the highest recorded weekly mortality rate among individuals infected with SARS-CoV-2 reached 101,600 deaths worldwide in January 2021",

consider "In January 2021, the weekly mortality rate from COVID-19 peaked at 101,600 deaths globally."

Use Subheadings: Introduce subheadings such as "Impact of COVID-19", "Challenges in Testing", and "Study Objectives" to organize content better.

Detailed Definitions: Provide a brief explanation of terms like 'repeated positivity' and 'internal references (IRs)' when they are first mentioned.

Highlight Novelty: Emphasize the unique aspects of the study early on, such as the use of RPP30 as a novel reference gene for assessing sample quality.

METHOD

Strengths

Detailed Description: The methods section provides comprehensive details about the sample collection, testing protocols, and various factors affecting sample quality.

Double-Blind Design: The use of a double-blind mode for sample collection increases the reliability and reduces bias in the study.

Control Experiments: Multiple control experiments addressing different stages of the testing process are well-documented, enhancing the robustness of the study.

Areas for Improvement (According to CONSORT Guidelines)

Participant Flow and Recruitment: The methods should include a detailed description of participant flow, including how many were screened, eligible, and included in the analysis. A CONSORT flow diagram would be beneficial.

Sample Size Calculation: There is no mention of a sample size calculation to justify the number of samples collected. Including this would strengthen the validity of the study's conclusions.

Randomization and Blinding: While the double-blind design is mentioned, the methods lack details on how randomization was performed. Clarifying the randomization process would enhance the study's transparency.

Interventions: The specific procedures for each intervention should be described in more detail, including any deviations from the protocol and how they were handled.

Outcomes: Clearly define primary and secondary outcomes. Specify the primary outcome measure, its assessment methods, and any thresholds used.

Statistical Methods: The statistical analysis section should include methods for handling missing data and conducting sensitivity analyses. Clarifying these methods would provide a complete picture of the analytical approach.

Specific Suggestions

Flow Diagram: Include a CONSORT flow diagram to illustrate the participant flow through each stage of the study.

Sample Size Justification: Add a section explaining the sample size calculation and the assumptions used.

Randomization Details: Provide a detailed description of the randomization process and how blinding was maintained.

Detailed Interventions: Expand on the procedural details for interventions, including any specific protocols followed.

Outcome Measures: Clearly define and justify the primary and secondary outcome measures, and describe how they were assessed.

Handling Missing Data: Outline methods for handling missing data and any sensitivity analyses conducted to ensure the robustness of the results.

Confounding Factors: Consideration of potential confounding factors and how they are controlled or adjusted for in the analysis should be included.

RESULTS

Strengths

Detailed Data Presentation: The results section provides comprehensive data on various aspects of the study, including overall sampling quality, effectiveness of RPP30, interference effects, and the impact of individual and non-sampler factors.

Statistical Analysis: Appropriate statistical methods are used to analyze the data, and significant results are clearly presented.

Use of Figures and Tables: The use of figures (e.g., Figure 2A-E, Figure 3A-G) and tables (e.g., Table 1, Table 2) helps in visualizing the data and understanding the results better.

Areas for Improvement (According to CONSORT Guidelines)

Flow of Participants: The section lacks a clear description of the flow of participants through each stage of the study, including the number of participants at each stage. Adding a CONSORT flow diagram would help clarify this.

Primary and Secondary Outcomes: Clearly define and distinguish between primary and secondary outcomes. This would help in understanding the main focus of the study and the additional findings.

Missing Data: Address how missing data were handled in the analysis. This is important to understand the completeness and reliability of the data.

Specific Suggestions

Participant Flow: Include a CONSORT flow diagram to illustrate the number of participants at each stage, including recruitment, participation, and analysis.

Outcome Definitions: Clearly define the primary and secondary outcomes, including how they were measured and analyzed.

Handling Missing Data: Provide a description of how missing data were handled, including any imputation methods used.

DISCUSSION

Strengths

Comprehensive Analysis: The discussion provides a thorough analysis of the results, including explanations for observed variances and correlations.

Practical Implications: The section discusses practical implications and potential improvements in sampling quality and testing reliability.

Contextual Relevance: The discussion effectively places the study's findings within the broader context of SARS-CoV-2 diagnostic challenges.

Areas for Improvement (According to CONSORT Guidelines)

Limitations: The section should explicitly address the limitations of the study, including potential biases and limitations in the methodology.

Generalizability: Discuss the generalizability of the findings to other settings, populations, and circumstances.

Future Research: Suggest directions for future research to address unanswered questions and further validate the study's findings.

Balanced Interpretation: Ensure a balanced interpretation of the results, avoiding overemphasis on positive findings and acknowledging any negative or unexpected results.

Consistency with Objectives: Clearly link the discussion back to the study's initial objectives and hypotheses to ensure consistency.

Specific Suggestions

Limitations Section: Add a paragraph explicitly discussing the study's limitations, such as potential sampling biases, the impact of missing data, and any constraints in the study design.

Generalizability: Include a discussion on how the findings might apply to other populations and settings, and whether the results can be generalized beyond the study sample.

Future Research Directions: Suggest specific areas for future research that could build on the current findings, such as exploring other potential biomarkers or testing methods.

Balanced Interpretation: Ensure that both the strengths and weaknesses of the study are discussed, providing a more balanced interpretation of the results.

Link to Objectives: Revisit the study's objectives and hypotheses in the discussion to ensure that the conclusions drawn are clearly connected to the initial aims of the study.

REFERENCES

Specific Suggestions

Formatting: Review the formatting of each reference to ensure consistency. For example, check for uniformity in author names, journal titles, volume and issue numbers, and page ranges.

Cross-Referencing: Ensure that all in-text citations match the references listed in the reference section. Cross-reference each citation to avoid omissions or incorrect references.

Prioritize Primary Research: Where secondary sources are cited, check if primary research articles are available and cite them directly if applicable.

Diverse Perspectives: Consider including more references that discuss potential biases, limitations, and differing results related to SARS-CoV-2 nucleic acid testing to provide a more comprehensive overview.

Dear Editor

The study is noteworthy because it is multicenter and has many cases. However, detailed information is required for false positivity. Also, I could not fully understand the ages of the patients. Was there standardization in the tests for under 2, 2-5, 5-12, 12-18 and 18 and over? I could not understand the age section. Was there a difference between lower respiratory tract Covid infection and upper respiratory tract COVID infection?

Best regards.

Dr Fatma Tuğba Çetin

Dear Editor

The study is noteworthy because it is multicenter and has many cases. However, detailed information is required for false positivity. Also, I could not fully understand the ages of the patients. Was there standardization in the tests for under 2, 2-5, 5-12, 12-18 and 18 and over? I could not understand the age section. Was there a difference between lower respiratory tract Covid infection and upper respiratory tract COVID infection?

Best regards.

Dr Fatma Tuğba Çetin

August 31, 2024

Dear editor and reviewers,

We wish to re-submit our revised manuscript titled "Exploring the Causes of Variability in Quality of Oropharyngeal Swab Sampling for SARS-CoV-2 Nucleic Acid Testing and Proposed Improvement Measures: A Multicenter, Double-Blind Study". The Manuscript ID is Spectrum01567-24.

We thank you for your thoughtful suggestions and insights. The manuscript has benefited from these helpful comments, and we look forward to working with you to move this manuscript closer to publication.

The manuscript has been rechecked and the necessary changes have been made in accordance with the your suggestions. We hope these meet the approval of you. These corrections do not influence the content and framework of the paper. The point-by-point responses to all comments as follows.

Editor's comments:

1. The authors need to clarify what is RPP30 and give the full term of RPP30

Response: This modification is indeed necessary. We have added a detailed introduction of RPP30, including its full name, in the background section of the manuscript. We have provided reasons and evidence for why RPP30 can be used as an internal reference gene in PCR testing (page7 line121-129 of marked-up manuscript).

2. The manuscript provides confusing information that may mislead readers to assume SARS-CoV-2 PCRs are quantitative, which is actually qualitative. Please double check

Response: Thank you for your questions. Clear explanations are crucial for understanding the manuscript. We have thoroughly reviewed the manuscript and found that only the section 'Verifying the Effectiveness of RPP30 in Evaluating Sampling Quality' involves SARS-CoV-2 PCR results. The main aim of this section is to validate the capability of RPP30 to represent sampling quality. We have established fixed standards for determining true positives, true negatives, and false negatives, as described in the original manuscript, and these determinations are not based on the Ct values of SARS-CoV-2 RNA alone. Interestingly, after selecting true positive samples, we observed a good linear relationship between the Ct values of RPP30 and two SARS-CoV-2 gene fragments. This suggests that in these isolated positive patients, the viral load consistency is strong, and the Ct values of PCR are influenced by the amount of epithelial cells in the collected samples. Thus, this phenomenon indirectly supports the effectiveness of RPP30 in evaluating sample quality.

3. Line 71-72: "These outbreaks continue to pose a serious risk of morbidity and mortality, among the elderly and immunocompromised individuals". More references should be added, with following as examples:

Role of Host Immune and Inflammatory Responses in COVID-19 Cases with Underlying Primary Immunodeficiency: A Review. J Interferon Cytokine Res. 2020 Dec;40(12):549-554. doi: 10.1089/jir.2020.0210. PMID: 33337932; PMCID: PMC7757688.

Response: Thanks for your attention to detail and for providing the literature, we have added citations.

4. Line 63-64: "The global population has been significantly impacted by the COVID-19 pandemic, which is caused by the SARS-CoV-2 virus, commonly known as the coronavirus". The authors should cite more references, with the following as example:

Clinical significance of measuring serum cytokine levels as inflammatory biomarkers in adult and pediatric COVID-19 cases: A review. *Cytokine*. 2021 Jun;142:155478. doi: 10.1016/j.cyto.2021.155478. Epub 2021 Feb 23. PMID: 33667962; PMCID: PMC7901304.

The Brief Case: Ventilator-Associated *Corynebacterium accolens* Pneumonia in a Patient with Respiratory Failure Due to COVID-19. *J Clin Microbiol*. 2021 Aug 18;59(9):e0013721. doi: 10.1128/JCM.00137-21. Epub 2021 Aug 18. PMID: 34406882; PMCID: PMC8372998.

Response: These documents are very helpful to improve the readability and completeness of this article, and we have added them.

5. Line 104-107: "Currently, COVID-19 and other virus test kits commonly utilize positive and negative standards to ensure experimental accuracy. However, these methods cannot verify the sample collection quality, such as whether there is a sufficient quantity of epithelial cells in the sample." It would be better to explain what "other virus" means here and add more examples and references to support this statement. For example, mpox PCR testing heavily relies on quality samples. This is also true for HSV and human parechovirus (HPeV) testing. The authors are suggested to add this into discussion. Some suggested references are listed here for the authors to consider:

Mpox (Monkeypox) Virus and Its Co-Infection with HIV, Sexually Transmitted Infections, or Bacterial Superinfections: Double Whammy or a New Prime Culprit? *Viruses*. 2024 May 15;16(5):784. doi: 10.3390/v16050784. PMID: 38793665; PMCID: PMC11125633.

The laboratory diagnosis of herpes simplex virus infections. *Can J Infect Dis Med Microbiol*. 2005 Mar;16(2):92-8. doi: 10.1155/2005/318294. PMID: 18159535; PMCID: PMC2095011.

Optimization and evaluation of a novel real-time RT-PCR test for detection of parechovirus in cerebrospinal fluid. *J Virol Methods*. 2019 Oct;272:113690. doi: 10.1016/j.jviromet.2019.113690. Epub 2019 Jul 5. PMID: 31283959.

Response: Adding specific examples and references to other viruses can assist in understanding and reading this article. Thank you for your suggestions, we have added them.

Reviewer #2 (Comments for the Author):

The study is noteworthy because it is multicenter and has many cases. However, detailed information is required for false positivity. Also, I could not fully understand the ages of the patients. Was there standardization in the tests for under 2, 2-5, 5-12, 12-18 and 18 and over? I could not understand the age section. Was there a difference between lower respiratory tract Covid infection and upper respiratory tract COVID infection? Best regards.

Response: Thank you for your valuable feedback and questions regarding our study.

1) Regarding false positives: The purpose of the section 'Verifying the Effectiveness of RPP30 in Evaluating Sampling Quality' is to validate the effectiveness of RPP30 in assessing sampling quality, specifically whether the epithelial cell count in the sample meets detection requirements. False positives are typically due to sample contamination or non-specificity of kit primers, which are not the focus of this study and cannot be identified through this reference. Additionally, when selecting true positive samples, we considered epidemiological history, symptoms, and results from dual kits and multiple repeated tests, which effectively minimizes the likelihood of false positives. In fact, COVID-19 testing primarily targets intracellular viruses, and since each swab is used by only one person, false positives are rare.

2) We apologize for the error in Table 1 where the age range 14-60 years was mistakenly listed as 14-16 years. The corrected version has been updated. In our study, we did not categorize ages so finely but only distinguished between three age groups. The conclusion emphasizes that staff should seek better cooperation when sampling children.

Reviewer #3 (Comments for the Author):

INTRODUCTION

Strengths

Comprehensive Overview: The introduction provides a thorough overview of the impact of the COVID-19 pandemic, covering mortality rates, ongoing risks, and the persistence of long COVID.

Focus on Testing Issues: The section highlights the critical role of nucleic acid testing and the challenges associated with it, such as false negatives and variability in test results.

Relevance: Discusses the practical implications of testing reliability and the importance of accurate sample collection.

Response: Thank you for your comments and recognition.

Areas for Improvement

Clarity and Conciseness: The introduction could benefit from more concise language. Some sentences are lengthy and could be split for better readability.

Structure and Flow: While the content is comprehensive, it can be better structured. Grouping related points together and using subheadings may enhance readability.

In-depth Explanation: Certain terms and concepts, such as 'repeated positivity' and 'internal references (IRs)', could be explained in more detail for readers who may not be familiar with these terms.

Citations and Evidence: Ensure that every claim, especially those related to the efficacy of specific genes like RPP30, is backed by citations to relevant studies or data.

Focus on Novel Contributions: The introduction should more clearly highlight the novel contributions of the study, setting it apart from existing literature.

Response: Thank you for your very helpful suggestions, and I will make modifications according to the

suggestions. Details are in the Specific Suggestions section below.

Specific Suggestions

Simplify Language: For example, instead of "During the initial three years of the COVID-19 pandemic, the highest recorded weekly mortality rate among individuals infected with SARS-CoV-2 reached 101,600 deaths worldwide in January 2021", consider "In January 2021, the weekly mortality rate from COVID-19 peaked at 101,600 deaths globally."

Response: Thank you for your very helpful suggestions. I made modifications according to the suggestions.

Use Subheadings: Introduce subheadings such as "Impact of COVID-19", "Challenges in Testing", and "Study Objectives" to organize content better.

Response: That sounds like a great improvement! Adding a focused subsection can really enhance the clarity and structure of your manuscript.

Detailed Definitions: Provide a brief explanation of terms like 'repeated positivity' and 'internal references (IRs)' when they are first mentioned.

Response: Indeed, adding this explanation is helpful for understanding the entire text. We have included the explanation in the relevant sections.

Highlight Novelty: Emphasize the unique aspects of the study early on, such as the use of RPP30 as a novel reference gene for assessing sample quality.

Response: Thank you. We have improved the final paragraph of the introduction (page7, line135-139 of marked-up manuscript).

METHOD

Strengths

Detailed Description: The methods section provides comprehensive details about the sample collection, testing protocols, and various factors affecting sample quality.

Double-Blind Design: The use of a double-blind mode for sample collection increases the reliability and reduces bias in the study.

Control Experiments: Multiple control experiments addressing different stages of the testing process are well-documented, enhancing the robustness of the study.

Response: Thank you for affirming this section.

Areas for Improvement (According to CONSORT Guidelines)

Participant Flow and Recruitment: The methods should include a detailed description of participant flow, including how many were screened, eligible, and included in the analysis. A CONSORT flow diagram would be beneficial.

Sample Size Calculation: There is no mention of a sample size calculation to justify the number of samples collected. Including this would strengthen the validity of the study's conclusions.

Randomization and Blinding: While the double-blind design is mentioned, the methods lack details on how randomization was performed. Clarifying the randomization process would enhance the study's transparency.

Interventions: The specific procedures for each intervention should be described in more detail, including any deviations from the protocol and how they were handled.

Outcomes: Clearly define primary and secondary outcomes. Specify the primary outcome measure, its assessment methods, and any thresholds used.

Statistical Methods: The statistical analysis section should include methods for handling missing data and conducting sensitivity analyses. Clarifying these methods would provide a complete picture of the analytical approach.

Response: Thank you very much for these suggestions; we completely agree with them. Detailed point-point explanations are provided below.

Specific Suggestions

Flow Diagram: Include a CONSORT flow diagram to illustrate the participant flow through each stage of the study.

Response: The CONSORT flow diagram was indeed not detailed enough; we have revised it.

Sample Size Justification: Add a section explaining the sample size calculation and the assumptions used.

Response: This information helps enhance the statistical rigor. We have added the sample size calculation process and parameter assumptions to the 'Sample Origin and Features' section (page8, line145-149 of marked-up manuscript).

Randomization Details: Provide a detailed description of the randomization process and how blinding was maintained.

Response: In the final paragraph of the 'Sample Origin and Features' section, we have expanded on the details of the random double-blind procedures (page9-10, line179-186 of marked-up manuscript).

Detailed Interventions: Expand on the procedural details for interventions, including any specific protocols followed.

Response: Indeed, this greatly enhances the rigor and scientific nature of the research. We have expanded on the details of the measures taken in the relevant sections. Additionally, it has been added to the discussion section that each unit's intervention measures should be tailored to its specific characteristics (page12, line231-238 of marked-up manuscript).

Outcome Measures: Clearly define and justify the primary and secondary outcome measures, and describe how they were assessed.

Response: After further validating the specificity and sensitivity of RPP30 in assessing sample epithelial cell counts and sample quality, we systematically investigated factors that might affect PCR results, such as RNA extraction stages and sample handling methods. The results of each experiment were determined based on the

analysis of differences in RPP30 internal reference Ct values. The main conclusion of the study is that the primary factor affecting sampling quality is the non-compliance of the sampler's collection process. Through linear regression analysis, we found that objective factors related to the sampler, such as job position, salary type, and sampling speed, might influence the sampling process. This represents our secondary conclusion. To improve the logical flow of the article, each section of the methods was enhanced to clarify the purpose of that part of the research and its significance within the overall study.

Handling Missing Data: Outline methods for handling missing data and any sensitivity analyses conducted to ensure the robustness of the results.

Response: 1) Handling Missing Data: During the COVID-19 pandemic, the volume of available samples was enormous, and we only needed to randomly select a subset that met the experimental and statistical requirements. For the prospective experiments, the sample size required was relatively small, and if RPP30 values were missing, we would repeat the experiment. In the section "Verifying the Effectiveness of RPP30 in Evaluating Sampling Quality," missing values could affect the determination of positive, negative, and false-negative samples, so we will exclude these samples. 2) Sensitivity Analysis: For the analysis of sample quality differences across 10 hospitals, this was observed as a phenomenon, and due to the large sample size and units, no additional hospitals were selected for external validation. The section "Verifying the Effectiveness of RPP30" is actually a replication of our previous study (PMID: 32749593), thus serving as an external validation. For the prospective experiments, we conducted three repetitions of each experiment, and the results were confirmed to be stable. Regarding the main conclusion of this paper—sampling quality is primarily affected by the sampler—both subjective and objective factors are involved. We did not perform data splitting but conducted linear regression using mixed samples from the 10 hospitals. Combining analysis or internal and external validation are common methods, each with its advantages and disadvantages. Combining analysis is generally used for a comprehensive assessment of research results, especially when the goal is to determine the overall effect. It offers high statistical power, strong representativeness, and better reflection of overall population characteristics and trends.

Finally, to enhance the completeness of the article, we have added these explanations to appropriate sections of the manuscript.

Confounding Factors: Consideration of potential confounding factors and how they are controlled or adjusted for in the analysis should be included.

Response: Starting from the conclusion that sampling quality is primarily affected by the sampler at the time of sample collection, it is crucial to consider confounding factors such as differences in sample processing stages, instruments, and sampling methods. Our prospective experiments in the middle sections specifically addressed these confounding factors. Additionally, we conducted a retrospective analysis of other confounding factors such as patient heterogeneity, seasonal variations, and departmental differences, which led us to conclude that the sampler is the key factor. Of course, the entire process of COVID-19 nucleic acid testing involves many details, such as sample transportation, instability of different batches of test kits, statistical organization of PCR result data, sampling environment, weather, and public health policies. These factors often have a smaller impact or are difficult to control. For example, the test kits used in the experiment are widely applied and robust, and since the sampling period was relatively short, batch effects were minimized. Thank you for your professional advice. In the revised manuscript's limitations discussion, we

have included these unaddressed confounding factors.

RESULTS

Strengths

Detailed Data Presentation: The results section provides comprehensive data on various aspects of the study, including overall sampling quality, effectiveness of RPP30, interference effects, and the impact of individual and non-sampler factors.

Statistical Analysis: Appropriate statistical methods are used to analyze the data, and significant results are clearly presented.

Use of Figures and Tables: The use of figures (e.g., Figure 2A-E, Figure 3A-G) and tables (e.g., Table 1, Table 2) helps in visualizing the data and understanding the results better.

Response: Thank you for your feedback.

Areas for Improvement (According to CONSORT Guidelines)

Flow of Participants: The section lacks a clear description of the flow of participants through each stage of the study, including the number of participants at each stage. Adding a CONSORT flow diagram would help clarify this.

Primary and Secondary Outcomes: Clearly define and distinguish between primary and secondary outcomes. This would help in understanding the main focus of the study and the additional findings.

Missing Data: Address how missing data were handled in the analysis. This is important to understand the completeness and reliability of the data.

Response: Thank you for your suggestions. Some explanations have been placed in the methods section. Below are some additional details and responses.

Specific Suggestions

Participant Flow: Include a CONSORT flow diagram to illustrate the number of participants at each stage, including recruitment, participation, and analysis.

Response: The CONSORT flow diagram was indeed not detailed enough, so we have made revisions.

Outcome Definitions: Clearly define the primary and secondary outcomes, including how they were measured and analyzed.

Response: After further validating the specificity and sensitivity of RPP30 in assessing sample epithelial cell counts and sample quality, we systematically investigated factors that might affect PCR results, such as RNA extraction stages and sample handling methods. The results of each experiment were determined based on the analysis of differences in RPP30 internal reference Ct values. The main conclusion of the study is that the primary factor affecting sampling quality is the non-compliance of the sampler's collection process. Through linear regression analysis, we found that objective factors related to the sampler, such as job position, salary type, and sampling speed, might influence the sampling process. This represents our secondary conclusion. To improve the logical flow of the article, each section of the methods was enhanced to clarify the purpose of

that part of the research and its significance within the overall study.

Handling Missing Data: Provide a description of how missing data were handled, including any imputation methods used.

Response: Handling Missing Data: During the COVID-19 pandemic, the volume of available samples was enormous, and we only needed to randomly select a subset that met the experimental and statistical requirements. For the prospective experiments, the sample size required was relatively small, and if RPP30 values were missing, we would repeat the experiment. In the section "Verifying the Effectiveness of RPP30 in Evaluating Sampling Quality," missing values could affect the determination of positive, negative, and false-negative samples, so we will exclude these samples (page8, line148-149 of marked-up manuscript).

DISCUSSION

Strengths

Comprehensive Analysis: The discussion provides a thorough analysis of the results, including explanations for observed variances and correlations.

Practical Implications: The section discusses practical implications and potential improvements in sampling quality and testing reliability.

Contextual Relevance: The discussion effectively places the study's findings within the broader context of SARS-CoV-2 diagnostic challenges.

Response: Thank you for your inquiry.

Areas for Improvement (According to CONSORT Guidelines)

Limitations: The section should explicitly address the limitations of the study, including potential biases and limitations in the methodology.

Generalizability: Discuss the generalizability of the findings to other settings, populations, and circumstances.

Future Research: Suggest directions for future research to address unanswered questions and further validate the study's findings.

Balanced Interpretation: Ensure a balanced interpretation of the results, avoiding overemphasis on positive findings and acknowledging any negative or unexpected results.

Consistency with Objectives: Clearly link the discussion back to the study's initial objectives and hypotheses to ensure consistency.

Response: Thank you very much for these suggestions; we completely agree with them. Detailed point-point explanations are provided below.

Specific Suggestions

Limitations Section: Add a paragraph explicitly discussing the study's limitations, such as potential sampling biases, the impact of missing data, and any constraints in the study design.

Response: This is indeed necessary. After our discussion, we have added a limitations section (page22,

line442-446 of marked-up manuscript), which is very helpful for improving the completeness of the article. Given the numerous and complex details of nucleic acid testing, it is challenging to address all aspects, and some factors may even be uncontrollable or unnoticed. Additionally, the characteristics affecting sampler work quality certainly include more than those we analyzed, and this may require managers to further review and refine the details.

Generalizability: Include a discussion on how the findings might apply to other populations and settings, and whether the results can be generalized beyond the study sample.

Response: This study collected a large amount of sample data from 10 hospitals, and the sample quality issues observed are highly representative. After thorough screening of multiple factors in nearly all key stages of nucleic acid testing, it was found that sampler-related factors have the greatest impact, which is both stable and noteworthy. The significant improvement in sample quality following the warnings not only provides valuable insights for developing improvement measures but also serves as a reverse validation of the main conclusions, reflecting the robustness of the conclusions and their potential value for broader application. This discussion has also been added to the discussion section. We have included some of this in the discussion section (page22, line435-442 of marked-up manuscript).

Future Research Directions: Suggest specific areas for future research that could build on the current findings, such as exploring other potential biomarkers or testing methods.

Response: This study used an *RPP30* internal reference kit to analyze factors affecting sampling quality. However, there are many meaningful areas for further research to address the limitations and deficiencies of this study. For example, exploring the impact of storage time at different temperatures, the effect of different transportation methods on results, and how the positive and negative quality controls of different kits influence the cutoff values for nucleic acid positivity. Additionally, further attention and continuous improvement are needed in the management of nucleic acid testing staff, including various methods, approaches, and training details (page23, line447-450 of marked-up manuscript).

Balanced Interpretation: Ensure that both the strengths and weaknesses of the study are discussed, providing a more balanced interpretation of the results .

Response: This is crucial for enhancing the objectivity of the article. Based on a multi-center, double-blind, randomized experiment, this study found that sampler-related factors are critically important in determining sampling quality and has identified and examined numerous other confounding factors that could potentially interfere with results during nucleic acid collection. The research conclusions are robust, reliable, and have broader implications. However, given the many and complex details of the nucleic acid testing process, it is impossible to account for all confounding factors, and sampler-related factors may only be one of the significant influences on the results. Additionally, the collection of covariates related to sampler characteristics is relatively limited; for instance, the heterogeneity in training quality and duration across different hospitals may also be considerable. Although enhancing management and policy formulation can improve sample quality to some extent, the screening and improvement of many objective factors, and the development of policies specific to each unit, require more attention from managers.

Link to Objectives: Revisit the study's objectives and hypotheses in the discussion to ensure that the

conclusions drawn are clearly connected to the initial aims of the study.

Response: Thank you for your reminder. The original intention of our study was to investigate whether there are significant quality issues in samples during the widespread nucleic acid testing and to identify the main factors contributing to these quality issues and potential improvements. The entire presentation of the main text revolves around this chain of inquiry. We have revised the document to make the presentation more organized and easier to understand.

REFERENCES

Specific Suggestions

Formatting: Review the formatting of each reference to ensure consistency. For example, check for uniformity in author names, journal titles, volume and issue numbers, and page ranges.

Response: Thank you for the reminder. We have conducted a thorough review.

Cross-Referencing: Ensure that all in-text citations match the references listed in the reference section. Cross-reference each citation to avoid omissions or incorrect references.

Response: Yes, after thorough review, we have ensured that the references are complete and accurate.

Prioritize Primary Research: Where secondary sources are cited, check if primary research articles are available and cite them directly if applicable.

Response: Thank you. We have adjusted some of the references to ensure that they are original research studies.

Diverse Perspectives: Consider including more references that discuss potential biases, limitations, and differing results related to SARS-CoV-2 nucleic acid testing to provide a more comprehensive overview.

Response: To present a more objective and multidimensional discussion on the complexity of SARS-CoV-2 nucleic acid testing, it is important to reference a diverse range of literature. This is crucial for establishing the foundational logic of the study. We have collected additional relevant literature to enhance the discussion and citations (citation 36).

Thank you again for your effort in our article, and thank you very much for your consideration. We hope that the revision is acceptable and look forward to hearing from you soon. Please contact us for any questions.

With kind regard,

Yours sincerely,

Zhenhua Zhang Prof

Department of Infectious Diseases, Clinical Virus Research Institute, The Second Affiliated Hospital of Anhui Medical University.

Address: Furong Road 678, Hefei 230601, Anhui, China.

Email: zzh1974cn@163.com

Re: Spectrum01567-24R1 (Exploring the Causes of Variability in Quality of Oropharyngeal Swab Sampling for SARS-CoV-2 Nucleic Acid Testing and Proposed Improvement Measures: A Multicenter, Double-Blind Study)

Dear Prof. Zhenhua Zhang:

Thank you for the privilege of reviewing your work. The Editor appreciates your effort to improve the quality of the manuscript. Below you will find my comments:

Editor's comments

ref23 should be "Liu BM, Rakhmanina NY, Yang Z, Bukrinsky MI. Mpox (Monkeypox) Virus and Its Co-Infection with HIV, Sexually Transmitted Infections, or Bacterial Superinfections: Double Whammy or a New Prime Culprit? *Viruses*. 2024;16(5):784."

Please return the manuscript within 7 days; if you cannot complete the modification within this time period, please contact me. If you do not wish to modify the manuscript and prefer to submit it to another journal, notify me immediately so that the manuscript may be formally withdrawn from consideration by Spectrum.

Revision Guidelines

Sincerely,
Benjamin Liu
Editor
Microbiology Spectrum

September 9th, 2024

Dear editor and reviewers,

We wish to re-submit our revised manuscript titled "Exploring the Causes of Variability in Quality of Oropharyngeal Swab Sampling for SARS-CoV-2 Nucleic Acid Testing and Proposed Improvement Measures: A Multicenter, Double-Blind Study". The Manuscript ID is Spectrum01567-24R1.

We thank you for your thoughtful suggestions and insights. The manuscript has benefited from these helpful comments, and we look forward to working with you to move this manuscript closer to publication.

The manuscript has been rechecked and the necessary changes have been made in accordance with the your suggestions. We hope these meet the approval of you. These corrections do not influence the content and framework of the paper. The point-by-point responses to all comments as follows.

Editor's comments:

ref23 should be "Liu BM, Rakhmanina NY, Yang Z, Bukrinsky MI. Mpox (Monkeypox) Virus and Its Co-Infection with HIV, Sexually Transmitted Infections, or Bacterial Superinfections: Double Whammy or a New Prime Culprit? *Viruses*. 2024;16(5):784."

Response: Thank you for your careful check. We are sorry that the information in reference 23 is not comprehensive, and we have revised it.

Thank you again for your effort in our article, and thank you very much for your consideration. We hope that the revision is acceptable and look forward to hearing from you soon. Please contact us for any questions.

With kind regard,

Yours sincerely,

Zhenhua Zhang Prof

Department of Infectious Diseases, Clinical Virus Research Institute, The Second Affiliated Hospital of Anhui Medical University.

Address: Furong Road 678, Hefei 230601, Anhui, China.

Email: zzh1974cn@163.com

Re: Spectrum01567-24R2 (Exploring the Causes of Variability in Quality of Oropharyngeal Swab Sampling for SARS-CoV-2 Nucleic Acid Testing and Proposed Improvement Measures: A Multicenter, Double-Blind Study)

Dear Prof. Zhenhua Zhang:

Your manuscript has been accepted, and I am forwarding it to the ASM production staff for publication. Your paper will first be checked to make sure all elements meet the technical requirements. ASM staff will contact you if anything needs to be revised before copyediting and production can begin. Otherwise, you will be notified when your proofs are ready to be viewed.

Sincerely,
Benjamin Liu
Editor
Microbiology Spectrum